# Powered Mobility Device Use and Developmental Change of Young Children with Cerebral Palsy

**DOI:** 10.3390/bs13050399

**Published:** 2023-05-10

**Authors:** Samuel W. Logan, Bethany M. Sloane, Lisa K. Kenyon, Heather A. Feldner

**Affiliations:** 1College of Health, Oregon State University, Corvallis, OR 97331, USA; sloaneb@oregonstate.edu; 2Department of Physical Therapy, Grand Valley State University, Grand Rapids, MI 49504, USA; kenyonli@gvsu.edu; 3Department of Rehabilitation Medicine, University of Washington, Seattle, WA 98195, USA; hfeldner@uw.edu

**Keywords:** cerebral palsy, disability, mobility, technology

## Abstract

Mobility is a fundamental human right and is supported by the United Nations and the ON Time Mobility framework. The purpose of this study was to understand the effect of a powered mobility intervention on developmental changes of children with cerebral palsy (CP). This study was a randomized, crossover clinical trial involving 24 children (12–36 months) diagnosed with CP or with high probability of future CP diagnosis based on birth history and current developmental status. Children received the Explorer Mini and a modified ride-on car in randomized order, each for 8 weeks. The Bayley Scales of Infant and Toddler Development—4th Edition was administered at baseline, mid-study, and end-of-study. Raw change scores were used for analysis. Total minutes of use per device was categorized as low or high use for analysis based on caregiver-reported driving diaries. Explorer Mini: The high use group exhibited significantly greater positive change scores compared to the low use group on receptive communication, expressive communication, and gross motor subscales (*p* < 0.05). Modified ride-on car: No significant differences between low and high use groups. Regardless of device, low use was associated with no significant developmental change and high use was associated with positive developmental changes. Mobility access is critical to maximize the development of children with CP and may be augmented by using powered mobility devices. Results may have implications for the development of evidence-based guidelines on dosage for powered mobility use.

## 1. Introduction

Mobility is a fundamental human right [1,2]. The United Nations supports this position of mobility equity as outlined in the Conventions on the Rights of Persons with Disabilities and the Rights of Children [3,4]. The ON Time Mobility framework further outlines children’s right to explore the environment, develop social relationships, and serve as active participants in co-creating experiences in their daily lives [2]. Mobility access is critical to maximize the development of children with neuromotor disabilities, including cerebral palsy (CP). Mobility may include traditional motor skill intervention as well as use of powered mobility devices such as motorized wheelchairs, modified ride-on cars, and the Explorer Mini, a mobility device designed specifically for toddlers.

Young children with CP demonstrate positive outcomes in mobility, development, and participation following a powered mobility intervention with a motorized wheelchair [5,6,7]. For example, young children with CP who used powered mobility devices exhibited increased mobility skills and independence [5,6,8,9], parent perceptions of social skills [6,10], receptive communication and self-care skills [7], sleep–wake patterns [10], and participation [11]. Children as young as 7 months old who have a range of motor abilities, including complex disabilities, have demonstrated successful engagement with powered mobility [7,12].

Despite these positive outcomes, there remain challenges to widespread adoption of powered mobility use, including child-related reasons such as perceived readiness based on age [13], cognitive, physical, or behavioral factors [14], and family or environment-related reasons such as lack of support, ability to transport the device, and home environment [14]. Another challenge includes caregiver perceptions that powered mobility use will interfere with their child’s motor development [11]. However, results of a randomized controlled trial suggest no significant differences in fine or gross motor skill between a powered mobility intervention group and a control group [7]. The ON Time Mobility framework does not provide readiness criteria for children to meet prior to consideration of powered mobility use but rather embraces a mobility rights perspective to advocate for multimodal access to mobility in many forms based on each child’s complex needs and environmental conditions [2]. Current recommendations indicate that powered mobility may be considered for children with disabilities as a means to explore mobility at ages and stages similar to their peers without disabilities, regardless of whether this is temporary, concurrent with gross motor skill intervention or as an anticipated long-term mobility solution [15].

Modified ride-on cars are an additional powered mobility option for young children with CP. Modified ride-on car use in young children with disabilities, including CP, is a feasible powered mobility option in home, hospital, and school settings and has been associated with positive activity and participation outcomes [16]. Modified ride-on cars include adaptation of commercially available, off-the-shelf, battery-operated toy cars. Modified ride-on cars can be adapted through installation of a large and easy to press activation switch on the steering wheel and customized seating support created from low-cost and readily available materials. The adapted switch usually includes an “all-or-nothing” activation mechanism where, once a child presses the switch, the car is turned on to its maximum speed until the switch is released, though, in some cases, potentiometers are also integrated to provide families with the ability to adjust speed. The total cost is about $200 for a modified ride-on car and modification supplies [17,18]. The do-it-yourself movement of modifying ride-on cars highlights a systemic gap in commercially available mobility technology for pediatric populations.

The Explorer Mini is a Food and Drug Administration cleared 510k medical device for young children 12–36 months old and was commercially released in March 2020. The Explorer Mini is activated with a midline joystick that provides proportional speed control. Other features include its zero-degree turning radius, five speed options, and the ability to be used in either a seated or standing position. A recent study of the Explorer Mini included 33 children 6–35 months old, 12 of which were diagnosed with CP. Results established initial feasibility for young children to successfully use the joystick for mobility and the observation that they appeared to enjoy the experience during a single driving session [19].

The current study extends previous work in three ways. First, this study addresses the potential effect of low and high device use on developmental outcomes in young children diagnosed with CP or with a high probability of future CP diagnosis. Previous intervention studies have examined powered mobility use and the onset of mobility skills [8,9] or generally reported use and developmental change without an interpretation of potential dosage effects [7]. Similarly, modified ride-on car use is highly variable, and, often, low adherence to use recommendations have been reported [16]. There are no studies with the Explorer Mini that report device use beyond a single driving session. Further, Permobil, manufacturer of the Explorer Mini, recently released “A Guideline for Introducing Powered Mobility to Infants and Toddlers.” [20]. The guideline acknowledges there are no specific recommendations for dosage of powered mobility use based on a lack of available evidence and the individually variable needs and abilities of many young drivers. Thus, the current study may have implications for the development of evidence-based guidelines for powered mobility use.

Second, this study examines children’s individual pathways from device use to developmental outcomes. Previous powered mobility intervention studies have used single-subject and case series designs to provide rich descriptions of behavior and developmental change [16,21]. However, the current study extends this work by using a unique approach of examining a larger sample size to synthesize individual-level data into larger trends that have the potential to impact clinical practice.

Third, our study aims, design, and interpretation of results are grounded in dynamic systems theory (DST) [22]. Regardless of device type, powered mobility intervention research is typically not grounded in theoretical frameworks of motor development. There are three key principles of DST: complexity, continuity in time, and dynamic stability. These principles interact to encourage an individual’s path toward a developmental cascade of change over time [22]. Complexity relates to the synergistic and interconnectedness of multiple systems that interact together and are influenced by the convergence of individual, task, and environmental constraints that influence behaviors. In the context of DST, constraints do not refer to limitations or restrictions but are the holistic context of how multiple systems interact to facilitate behaviors. We recognize that child development is complex and may be influenced by children’s Gross Motor Function Classification System (GMFCS) level (individual constraint) and the use of the Explorer Mini and modified ride-on car (task constraints) in different family, home, neighborhood, and community spaces that present varying real-world situations of powered mobility use (environmental constraints). Continuity in time recognizes that change in functioning is dependent on the past, which influences the path toward future levels of functioning. Our study acknowledges continuity through examining children’s individual pathways of developmental change, thereby recognizing that each child is likely to experience their own unique trajectory dependent upon their previous developmental past. Dynamic stability regards behaviors as stable and flexible to varying degrees, depending upon the behavior and state of the system at a given point in time. Our study embraces that dynamic stability of developmental change may be influenced by the frequency of powered mobility device use between assessments of developmental domains.

The purpose of this study was to examine the effect of powered mobility use on the developmental changes of young children (12–36 months) diagnosed with CP or with a high likelihood of future CP diagnosis following separate 8-week use periods for the Explorer Mini and modified ride-on car. Children between 12 and 36 months of age were the focus of this study because early childhood is a critical developmental time and provision of powered mobility is not standard of practice at this age [13,14] despite previous positive research findings [7,8,9,10,12,16]. Therefore, the overall objective was to understand the effect of a powered mobility intervention on developmental changes. In the United States, there was no commercially available powered mobility device for children under 3 years of age until the Explorer Mini was released in 2020. Prior to the Explorer Mini, modified ride-on cars were popularized as a do-it-yourself powered mobility option for young children. Both devices were chosen for the current study because of their use for children 12–36 months as a powered mobility device for this population. In addition, there are cost and access differences between the devices. The Explorer Mini costs $2944 and requires a physician’s prescription for access. In contrast, a modified ride-on car costs $200–$400 and requires minimal technical skills for access. These factors contributed to the use of a randomized, crossover study design that included children using each device during an intervention period.

There were two aims of the current study. Aim 1: Compare the relationship between device use frequency (low and high use) of each device to change scores of Bayley-4 subscales (e.g., cognitive, receptive communication, expressive communication, fine motor, and gross motor). H1: *We hypothesized that change scores across all Bayley-4 subscales would be higher for the high use group compared to the low use group for both devices.* Aim 2: Describe children’s individual pathways of developmental change on Bayley-4 subscales considering device use and GMFCS levels. H2: *We hypothesized that high use would be associated with positive developmental changes for both devices.* H3: *We also hypothesized that low use would be associated with no developmental changes.* H4: *Lastly, we hypothesized that children classified at GMFCS Levels I, II, and III would exhibit more pathways to positive developmental change compared to children classified at GMFCS Levels IV and V.*

## 2. Materials and Method

This study was a randomized, crossover, multi-site clinical trial, and children received the Explorer Mini or a modified ride-on car in a randomized order, each for 8 weeks. There was no wash out period between devices since both devices are intended to support self-initiated mobility. This study is part of a larger study. Please see [23] for a published protocol with full methodological details.

### 2.1. Participants

Recruitment of potential participants was conducted through local physical, occupational, and early intervention agencies and clinics at each site (Washington, Oregon, and Michigan). Twenty-four children between 12 and 36 months of age diagnosed with CP or with high probability of future CP diagnosis based on birth history or current developmental status were included in this study. A high probability of future CP diagnosis was confirmed through caregiver report based on birth history or current developmental status, demonstrated delays of the onset of mobility, and receipt of therapeutic services. One family did not return the caregiver-reported driving diary about device use and were excluded from analysis, resulting in a final sample of 23 children for the current study. See Table 1 for demographic information.

### 2.2. Description of Devices

Explorer Mini. The Explorer Mini is commercially available and Food and Drug Administration cleared 510k medical device intended for use of children 12–36 months of age. The Explorer Mini includes a rechargeable, 12-volt battery, maximum speed of 1.5 mph, a 0-inch turning radius, five speed options, can be driven in sitting or standing positions, 35 lbs. weight limit, and is activated and steered through an omni-directional and proportional controlled joystick. See Figure 1.

Modified ride-on car. The Fisher Price Cars 3 Lil’ Lightning McQueen is commercially available and intended for use of children 12–36 months. The McQueen includes a rechargeable, 6-volt battery, maximum speed of 2 mph, a 37.5 inch turning radius, one speed option, can be used in the sitting position only, 40 lbs. weight limit, steered via a handheld steering wheel, and activated through an all-or-nothing switch pressed via a finger or thumb located on the steering wheel. Modifications included (a) replacing the small switch with an all-or-nothing adapted switch that is large (5-inch diameter), easy to press, and installed on the steering wheel, (b) the addition of a potentiometer to allow for variable speed control, and (c) customized and individual seating support based on each child’s positioning needs. See Figure 2.

### 2.3. Dependent Variables

Device use. Caregivers reported device use as minutes per driving session in a caregiver-reported driving diary. Several modified ride-on car studies recommended to families at least 20–30 min per day for 5 days per week of device use; however, actual device use is often low and highly variable [16]. Families in the current study were encouraged to incorporate the devices into their everyday routines and participated in two standardized check-in periods per device to encourage driving and identify/remove potential driving barriers; however, they were not provided specified device use recommendations. We created definitions of low and high use based on our research, clinical experience, and previous literature [16,24]. Low use was defined as 480 min or less across an 8-week period. This is equivalent to an average of 20 min per day for 3 days per week (i.e., 1 h per week or less, which is similar to dosage of early intervention services) [24]. High use was defined as 481 min or more across an 8-week period. Low and high use groups were used for data analysis.

### 2.4. Bayley Scales of Infant and Toddler Development—4th Edition (Bayley-4) [25]

The Bayley-4 is a norm-referenced standardized measure that was validated with a sample of children mostly without disabilities. Scaled and raw scores may be used to identify change over time, but scaled scores must be used to compare a child’s performance to their age-matched peers. However, in a heterogenous sample of children with disabilities, even in the presence of significant change in raw scores, scaled scores may remain steady or decline/decrease due to the inherent comparison against age-matched peers. Raw scores were used to calculate change scores for each subscale between each period of device use because of the sample and relatively short intervals between assessment (8 weeks). Raw scores and change scores more accurately reflect the presence or absence of change in our participants who served as their own controls, which considers our population (i.e., CP) heterogeneity. Further, caregivers reported that 14 of the 23 children (~61%) functioned at GMFCS Levels IV or V. These children are expected to develop at a decreased rate compared to other GMFCS levels, or compared to children with typical development who were the basis for the norm referenced scaled and standard scores in the Bayley-4 manual.

The Bayley-4 was administered at T0 (prior to any device use), T1 (after 8 weeks of first device use), and T2 (after 8 weeks of second device use) and included assessment of the cognitive, receptive communication, expressive communication, fine motor, and gross motor subscales. Change scores were calculated by subtracting the raw score at one time point from the raw score at another timepoint for each individual child. The percentage of change was calculated by dividing the change score by the raw score at first timepoint ×100. For example, a raw score of 57 at T0 and 70 at T1 would result in a change score of 13 (70–57), which represents 23% positive change (13/57 = 0.2280 × 100 = 23%). The magnitude of the percentage of change was defined as follows: stable (+/−9% or less), small change (+/−10–19%), moderate change (+/−20–29%), or large change (+/−30% or more). We used a conservative approach informed by our collective research and clinical experience to define magnitudes of percentage of change. The context of social validity, including the importance of the treatment effect, guided the classification of magnitudes [26,27]. Similar to a previous powered mobility study, the lowest level of change determined as meaningful was defined as at least 10% because this level of change may inform intervention planning [28].

### 2.5. Data Analysis

Aim 1: Compare the relationship between device use frequency (low and high use) of each device to change scores of Bayley-4 subscales. Non-parametric tests were used due to small sample size of groups and the violation of the assumption of normality (Shapiro–Wilk test; *p* < 0.05) for the receptive communication and gross motor change scores. The Mann–Whitney U test was used to compare two independent groups (low vs high use) on Bayley-4 change scores. Separate tests were conducted across Bayley-4 subscales (cognitive, receptive communication, expressive communication, fine motor, and gross motor) and across devices (Explorer Mini; modified ride-on car).

Aim 2: Describe children’s individual pathways of developmental change on Bayley-4 subscales considering device use and GMFCS levels. Visual analysis was used to narratively describe trends through a series of figures. Figures are presented that include children’s individual pathways of percentage change (stable, small, moderate, large) across Bayley-4 subscales (cognitive, receptive communication, expressive communication, fine motor, and gross motor), devices (Explorer Mini; modified ride-on car), use levels (low use; high use), and GMFCS Levels (I–III; IV–V).

## 3. Results

Aim 1: See Table 2 for descriptive information about device use. **Explorer Mini.** Expressive communication change scores of the high use group (Mdn = 4) were higher than those of the low use group (Mdn = 1.5). A Mann–Whitney U test indicated that this difference was statistically significant, *U(Nhigh use = 11*, *Nlow use = 12) = 32.00*, *z =* −2.1, *p =* 0.037. Receptive communication change scores of the high use group (Mdn = 4) were higher than those of the low use group (Mdn = 0). A Mann–Whitney U test indicated that this difference was statistically significant, *U(Nhigh use = 11*, *Nlow use = 12) = 15.50*, *z =* −3.1, *p <* 0.001. Gross motor change scores of the high use group (Mdn = 5) were higher than those of the low use group (Mdn = −0.5). A Mann–Whitney U test indicated that this difference was statistically significant, *U(Nhigh use = 11*, *Nlow use = 12) = 32.00*, *z =* −2.1, *p =* 0.037. **Modified ride-on car.** No significant differences in change scores resulted between low and high use groups.

Aim 2: A narrative description is provided for children’s individual pathways of developmental change on Bayley-4 subscale scores. See Figure 3, Figure 4, Figure 5, Figure 6 and Figure 7.

**Cognitive subscale. Explorer Mini: Low use (*n* = 12; 52%).** Children exhibited five of seven possible individual pathways from low use to developmental change. The most common pathway was low use to positive change (*n* = 6; 50%), including small (*n* = 1), moderate (*n* = 3), and large (*n* = 2). The next most common pathway was low use to stable (*n* = 4; 33%). The least common pathway was low use to negative change (*n* = 2; 17%), including small (*n* = 0), moderate (*n* = 2), and large (*n* = 0). Children with GMFCS I–III and IV–V appeared to show similar patterns. **High use (*n* = 11; 48%).** Children exhibited five of seven possible individual pathways from high use to developmental change. The most common pathway was high use to stable (*n* = 5; 46%). The next most common pathway was high use to positive change (*n* = 4; 36%), including small (*n* = 1), moderate (*n* = 1), and large (*n* = 2). The least common pathway was high use to negative change (*n* = 2; 18%), including small (*n* = 2). Children with GMFCS I–III and IV–V appeared to show similar patterns.

**Modified ride-on car: Low use (*n* = 18; 78%).** Children exhibited seven of seven possible individual pathways from low use to developmental change. The most common pathway was low use to stable (*n* = 8; 44%). The remaining two pathways were equal in commonality, including low use to negative change (*n* = 5; 28%), including small (*n* = 3), moderate (*n* = 1), and large (*n* = 1); and positive change (*n* = 5; 28%), including small (*n* = 2), moderate (*n* = 2), and large (*n* = 1). Children with GMFCS I–III and IV–V appeared to show similar patterns. **High use (*n* = 5; 22%).** Children exhibited three of seven possible individual pathways from high use to developmental change. The most common pathway was high use to positive change (*n* = 3; 60%), including small (*n* = 2) and large (*n* = 1). The next most common pathway was high use to negative change (*n* = 2; 40%), including small (*n* = 2). The least common pathway was high use to stable (*n* = 0; 0%). Children with GFMCS I–III all showed negative change while all children with GMFCS IV–V showed positive change.

**Receptive subscale. Explorer Mini: Low use (*n* = 12; 52%).** Children exhibited six of seven possible individual pathways from low use to developmental change. The most common pathway was low use to stable (*n* = 5; 42%). The next most common pathway was low use to positive change (*n* = 4; 33%), including small (*n* = 3) and moderate (*n* = 1). The least common pathway was low use to negative change (*n* = 3; 25%), including small (*n* = 1), moderate (*n* = 1), and large (*n* = 1). Children with GMFCS I–III and IV–V appeared to show similar patterns. **High use (*n* = 11; 48%).** Children exhibited four of seven possible individual pathways from high use to developmental change. The most common pathway was high use to stable (*n* = 6; 66%). The next most common pathway was high use to positive change (*n* = 5; 46%), including small (*n* = 2), moderate (*n* = 1), and large (*n* = 2). The least common pathway was high use to negative change (*n* = 0; 0%). Children with GMFCS I–III and IV–V appeared to show similar patterns.

**Modified ride-on car: Low use (*n* = 18; 78%).** Children exhibited four of seven possible individual pathways from low use to developmental change. The most common pathway was low use to stable (*n* = 10; 56%). The next most common pathway was low use to positive change (*n* = 6; 33%), including small (*n* = 1) and moderate (*n* = 5). The least common pathway was low use to negative change (*n* = 2; 11%), including small (*n* = 2). Children with GMFCS IV–V appeared to show more positive changes compared to GMFCS I–III. **High use (*n* = 5; 22%).** Children exhibited four of seven possible individual pathways from high use to developmental change. There were two most common pathways: high use to positive change (*n* = 2; 40%), including small (*n* = 2); and high use to negative change (*n* = 2; 40%), including small (*n* = 1) and moderate (*n* = 1). The least common pathway was high use to stable (*n* = 1; 20%). Children with GMFCS I–III and IV–V appeared to show similar patterns.

**Expressive subscale. Explorer Mini: Low use (*n* = 12; 52%).** Children exhibited five of seven possible individual pathways from low use to developmental change. The most common pathway was low use to positive change (*n* = 5; 42%), including small (*n* = 4) and moderate (*n* = 1). The next most common pathway was low use to stable (*n* = 4; 33%). The least common pathway was low use to negative change (*n* = 3; 25%), including small (*n* = 1) and large (*n* = 2). Children with GMFCS I–III and IV–V appeared to show similar patterns. **High use (*n* = 11; 48%).** Children exhibited three of seven possible individual pathways from high use to developmental change. The most common pathway was high use to positive change (*n* = 10; 91%), including small (*n* = 6) and large (*n* = 4). The next most common pathway was high use to stable (*n* = 1; 9%). The least common pathway was high use to negative change (*n* = 0; 0%). Children with GMFCS I–III and IV–V appeared to show similar patterns.

**Modified ride-on car: Low use (*n* = 18; 78%).** Children exhibited five of seven possible individual pathways from low use to developmental change. The most common pathway was low use to positive change (*n* = 7; 39%), including small (*n* = 6) and large (*n* = 1). The next most common pathway was low use to stable (*n* = 6; 33%). The least common pathway was low use to negative change (*n* = 5; 28%), including small (*n* = 2) and moderate (*n* = 3). Children with GMFCS I–III and IV–V appeared to show similar patterns. **High use (*n* = 5; 22%).** Children exhibited three of seven possible individual pathways from high use to developmental change. There were two most common pathways: high use to positive change (*n* = 2; 40%), including large (*n* = 2); and high use to stable (*n* = 2; 40%). The least common pathway was from high use to negative change, including small (*n* = 1; 20%). Children with GMFCS I–III and IV–V appeared to show similar patterns.

**Fine motor subscale. Explorer Mini: Low use (*n* = 12; 52%).** Children exhibited six of seven possible individual pathways from low use to developmental change. The most common pathway was low use to stable (*n* = 5; 42%). The next most common pathway was low use to negative change (*n* = 4; 33%), including small (*n* = 1), moderate (*n* = 2), and large (*n* = 1). The least common pathway was low use to positive change (*n* = 3; 25%), including small (*n* = 2) and moderate (*n* = 1). Children with GMFCS IV–V showed positive, stable, and negative change, while children with GMFCS I–III showed only stable or negative change. **High use (*n* = 11; 48%).** Children exhibited six of seven possible individual pathways from high use to developmental change. The most common pathway was high use to positive change (*n* = 6; 55%), including small (*n* = 1), moderate (*n* = 1), and large (*n* = 4). The next most common pathway was high use to negative change (*n* = 3; 27%), including small (*n* = 1) and moderate (*n* = 2). The least common pathway was high use to stable (*n* = 2; 18%). Children with GMFCS I–III and IV–V appeared to show similar patterns.

**Modified ride-on car: Low use (*n* = 18; 78%).** Children exhibited six of seven possible individual pathways from low use to developmental change. The most common pathway was low use to stable (*n* = 8; 44%). The next most common pathway was low use to positive change (*n* = 6; 33%), including small (*n* = 3) and large (*n* = 3). The least common pathway was low use to negative change (*n* = 4; 22%), including small (*n* = 1), moderate (*n* = 2), and large (*n* = 1). Children with GMFCS IV–V showed positive, stable, and negative change, while children with GMFCS I–III showed only stable or positive change. **High use (*n* = 5; 22%).** Children exhibited 3 of 7 possible individual pathways from high use to developmental change. The most common pathway was high use to positive change (*n* = 4; 80%), including small (*n* = 3) and large (*n* = 1). The next most common pathway was high use to stable (*n* = 1; 20%). The least common pathway was from high use to negative change (*n* = 0; 0%). Children with GMFCS I–III and IV–V appeared to show similar patterns.

**Gross motor subscale. Explorer Mini: Low use (*n* = 12; 52%).** Children exhibited five of seven possible individual pathways from low use to developmental change. The most common pathway was low use to negative change (*n* = 6; 50%), including small (*n* = 2), moderate (*n* = 3), and large (*n* = 1). The next most common pathway was low use to stable (*n* = 5; 42%). The least common pathway was low use to positive change (*n* = 1; 8%), including large (*n* = 1). Children with GMFCS I–III showed stable pathways compared to children with GMFCS IV–V, who tended to show negative changes. **High use (*n* = 11; 48%).** Children exhibited 4 of 7 possible individual pathways from high use to developmental change. The most common pathway was high use to positive change (*n* = 8; 73%), including small (*n* = 2), moderate (*n* = 1), and large (*n* = 5). The next most common pathway was high use to stable (*n* = 3; 27%). The least common pathway was high use to negative change (*n* = 0; 0%). Children with GMFCS IV–V always showed positive change, while children with GFMCS I–III showed stable pathways and positive change.

**Modified ride-on car: Low use (*n* = 18; 78%).** Children exhibited four of seven possible individual pathways from low use to developmental change. The most common pathway was low use to stable (*n* = 14; 78%). The next most common pathway was low use to positive change (*n* = 3; 17%), including small (*n* = 1) and large (*n* = 2). The least common pathway was low use to negative change (*n* = 1; 6%), including small (*n* = 1). Children with GMFCS I–III and IV–V appeared to show similar patterns. **High use (*n* = 5; 22%)**. Children exhibited two of seven possible individual pathways from high use to developmental change. The most common pathway was high use to stable (*n* = 3; 60%). The next most common pathway was high use to positive change (*n* = 2; 40%), including moderate (*n* = 2). The least common pathway was from high use to negative change (*n* = 0; 0%). Children with GMFCS IV–V showed stable pathways and positive change, while children with GMFCS I–III showed only stable pathways.

**Summary of individual pathways across all Bayley-4 domains.** See Table 3. Regardless of device, the most common pathway for low use was to stable (*n* = 69; 46%) and for high use to positive change (*n* = 46; 57.5%).

**Explorer Mini: Low use (*n* = 12; 52%).** There were 60 pathways recorded from low use to developmental change (positive, stable, negative). The most common pathway was low use to stable (*n* = 23; 38%). **High use (*n* = 11).** There were 55 pathways recorded from high use to developmental change (positive, stable, negative). The most common pathway was high use to positive change (*n* = 33; 60%).

**Modified ride-on car: Low use (*n* = 18).** There were 90 pathways recorded from low use to developmental change (positive, stable, negative). The most common pathway was low use to stable (*n* = 46; 51%). **High use (*n* = 5).** There were 25 pathways recorded from high use to developmental change (positive, stable, negative). The most common pathway was high use to positive change (*n* = 13; 52%).

## 4. Discussion

The purpose of this study was to examine the effect of powered mobility use on the developmental changes of young children diagnosed with CP or with a high likelihood of future CP diagnosis. Our first hypothesis was partially supported and stated that change scores across all Bayley-4 subscales would be higher for the high use group compared to the low use group for both devices. Findings indicate that high use of the Explorer Mini resulted in significantly greater change scores compared to low use on receptive communication, expressive communication, and gross motor domains. There were no significant differences between low use and high use of a modified ride-on car.

One potential explanation for our findings is that children’s high use of the Explorer Mini may have contributed to new exploratory experiences that resulted in increased, varied, and novel social interactions with caregivers, siblings, and/or others in the environment, which, in turn, facilitated changes in receptive communication and expressive communication. The powered mobility experiences may have motivated children to be more active and mobile outside of the device, contributing to advanced gross motor skills. A substantially higher percentage of children were in the high use group for the Explorer Mini (48%) compared to the modified ride-on car (22%). Further, there was a low percentage of children who were in the high use group for both devices (17%). These results are consistent with previous work that indicates variable duration and frequency of device use during an intervention period [16] that is likely due to several perceived barriers related to the caregiver, child, device, and environment [29,30]. Our study protocol attempted to address these issues through providing families with two standardized check ins during each 8-week period of device use. These check ins included time for the caregiver to ask questions and discuss perceived barriers, and for the research team to provide activity suggestions and facilitating strategies to encourage both device use and children’s learning. Future research is warranted to understand the exploratory experiences of young children during powered mobility device use to determine if and how these experiences contribute to developmental change, including communication skills.

Another potential explanation is the caregivers’ and children’s device preferences. There were 11 families in the high use group of the Explorer Mini. Based on our qualitative data, 8 out of 11 families indicated that both caregiver and child preferred the Explorer Mini compared to the modified ride-on car. It is possible the preference for the Explorer Mini both in quantitative (i.e., use) and qualitative ways influenced the frequency, quality, and type of opportunities children were provided to use the device that contributed to the observed developmental changes.

Lastly, a potential explanation for the findings is related to the functional difference in how the Explorer Mini and modified ride-on car are operated and used to navigate the environment. The Explorer Mini uses a joystick for activation of omni-directional steering, while the modified ride-on car uses an all-or-nothing and single switch for activation that is separate from steering control. In combination with high use, the joystick navigation of the Explorer Mini may have resulted in different mobility experiences that can at least partially explain the findings. To our knowledge, there are no research studies that directly compares children’s driving experiences of powered mobility devices that are activated through a single switch versus a joystick within home and community settings, and further research is warranted.

Our second and third hypotheses were supported and stated that, through visual analysis of individual pathways, high use would be associated with positive developmental changes and that low use would be associated with no developmental changes for both devices. The most common pathway from high use was to positive change for the Explorer Mini (60% of pathways) and modified ride-on car (53% of pathways). The most common pathway from low use was to no developmental change (i.e., stable) for the Explorer Mini (38% of pathways) and modified ride-on car (51% of pathways). Despite the common pathways, it is clear that this is not a hard and fast rule, and there are several factors that influence a child’s developmental trajectory that align with dynamic systems theory. Bi-directional interactions amongst individual (children’s previous developmental history and current GMFCS level), task (device preference and use), and environmental (settings of device use) constraints likely contributed in different ways to development change for each child in the current study. These results align with a classic study in motor development where researchers examined individual pathways in the development of the fundamental motor skill of throwing [31]. They found common pathways, yet there was variability in how the trunk, humerus, and forearm actions coordinate to produce throwing across trials and time. Analyses of different groups are important in research studies, but there is also value in examining individual data to understand the underlying patterns of change.

Our fourth hypothesis was not supported; it stated that, through visual analysis of individual pathways, children classified at GMFCS Levels I, II, and III would exhibit more pathways to positive developmental change compared to children classified at GMFCS Levels IV and V. On the cognitive subscale, all children at GMFCS Levels I, II, and III showed negative change while all children at GMFCS Levels IV and V showed positive change following modified ride-on car use. On the gross motor subscale, a mix of stable pathways and positive change were demonstrated across GMFCS levels and devices. There were no discernable trends on the receptive communication, expressive communication, and fine motor subscales. These findings were not dependent on high use of devices since there was a similar breakdown of children in the high use group for each device at GMFCS Levels I, II, III and GMFCS Levels IV and IV (45% and 55% respectively for high use of the Explorer Mini; 40% and 60% respectively for high use of a modified ride-on car). These results have important research and clinical applications. Children at GMFCS Levels IV and V are often excluded from powered mobility research trials due to safety and readiness concerns related to limited head, trunk, and limb control. Often, an inclusion or exclusion criteria is related to a child’s ability to sit with support as a requirement for study enrollment [19,32]. Our results clearly demonstrate that children at GMFCS Levels IV and V should be included in powered mobility research trials. In the current study, children at GMFCS Levels IV and V demonstrated the most frequent amount of positive change for certain Bayley-4 domains, further highlighting the clinical applicability of powered mobility intervention in this population.

It is important to acknowledge the limitations of the current study. First, our study is statistically powered for its primary aims [23]. However, the study is not statistically powered to examine differences amongst several subgroups such as device use (i.e., high, low), device type (i.e., Explorer Mini, modified ride-on car), and GMFCS Levels (i.e., I, II, II and IV, V). Nonetheless, this study is part of the largest powered mobility clinical trial to date, and the examination of individual pathways of developmental change provides new knowledge. Second, the current study included children from 12 to 36 months of age at the time of enrollment. Although this is a common age range of powered mobility research studies [7,16], it is important to note that any observed effects may have been influenced by an interaction of age, experience, and functional mobility. Third, the classification of low and high use of devices was based on caregiver-reported driving diaries. A recent study compared modified ride-on car use measured through objective tracking or caregiver diaries [32]. There were no significant differences between objective tracking and diaries on average session duration or total driving time. The authors noted that over- or under-reporting of use through diaries may have occurred, but researchers can reasonably expect that caregiver diaries accurately represent their child’s device use. The objective tracking used in previous work involves directly integrating hardware components into the electrical system of the modified ride-on car [32]. These components are not compatible with the Explorer Mini. There is a need for further sensor development and integration with powered mobility devices to understand how they are used in home and community spaces. This type of technology is readily available on traditional power chairs for adults, but this remains a salient need in the pediatric population. Fourth, in-depth information about other factors that may have contributed to the observed changes in the current study were not systematically measured and controlled for in analyses, such as the frequency, duration, and specific activities of other therapies received, interactions between caregivers and children during device use, participation in other play-based experiences, or any number of environmental circumstances such as the size and type of house or surrounding built environment of the neighborhood and community. Fifth, differences between the use amounts, device characteristics, and family preferences of devices makes it difficult to draw conclusions about specific factors that may have contributed to the observed developmental changes. Our results suggest more work is needed with separate groups assigned to each device and a standardized dosage to further understand effects of a powered mobility intervention. Lastly, Bayley-4 standard scores are not available for young children with CP across each GMFCS level. Therefore, it is unknown whether the magnitude of our observed changes in raw scores are expected for this population; however, it is unlikely, given the short time of 8 weeks between assessments. In addition, we used change scores relative to each child, so they served as their own control. Results of the current study should be interpreted with caution and may not be generalizable to all young children with CP.

## 5. Conclusions

In conclusion, mobility is a fundamental human right [1]. This is a position supported by the United Nations and the ON Time Mobility framework [2,3,4]. The multimodal principle of this framework advocates for children to have a range of technology options for mobility depending upon what works best for them based on an interaction of their individual and environmental constraints. Powered mobility is one mobility option for young children with CP. The results of the current study indicate the potential for positive developmental change following high use of a powered mobility device, and they recognize the variability of individual differences in children’s developmental trajectories and potentially differing responses to intervention.

## Figures and Tables

**Figure 1 behavsci-13-00399-f001:**
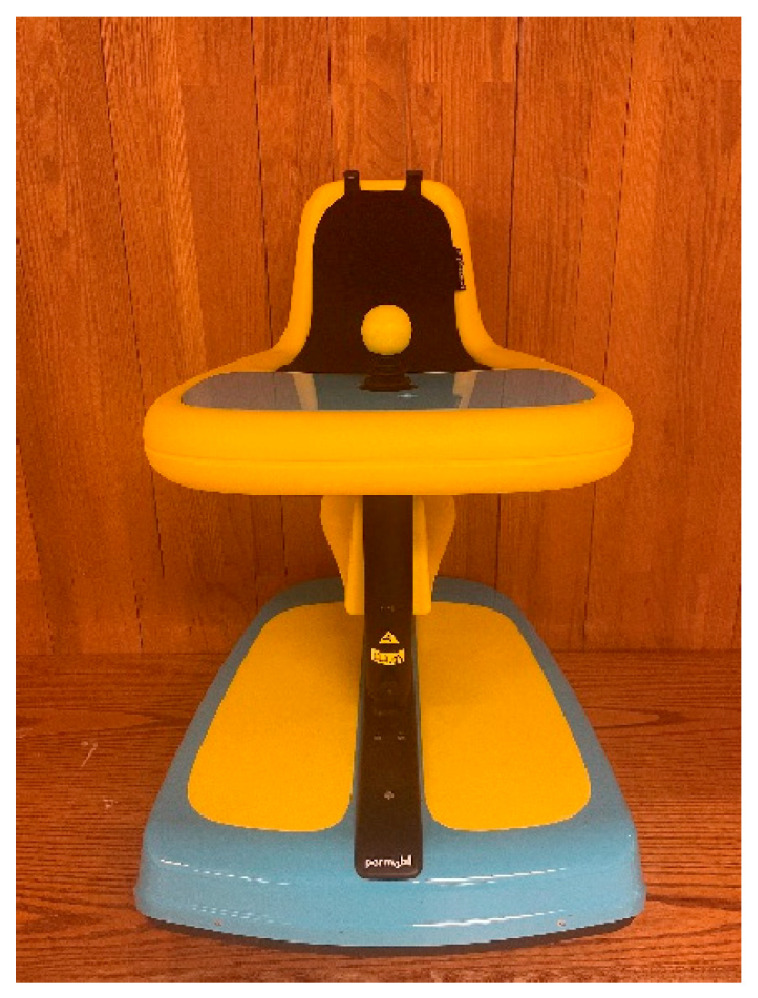
Photograph of the Explorer Mini.

**Figure 2 behavsci-13-00399-f002:**
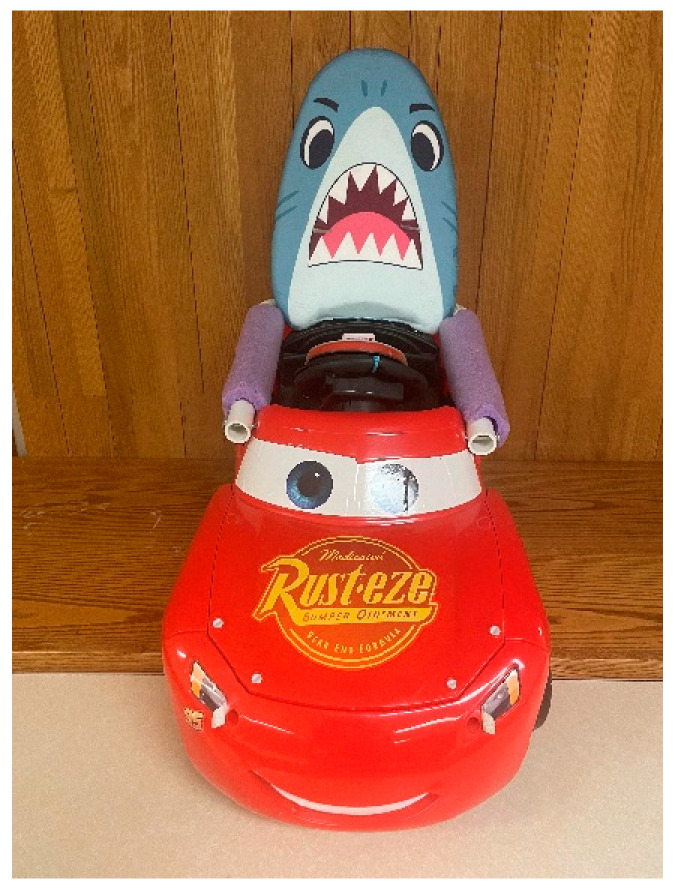
Photograph of the modified ride-on car.

**Figure 3 behavsci-13-00399-f003:**
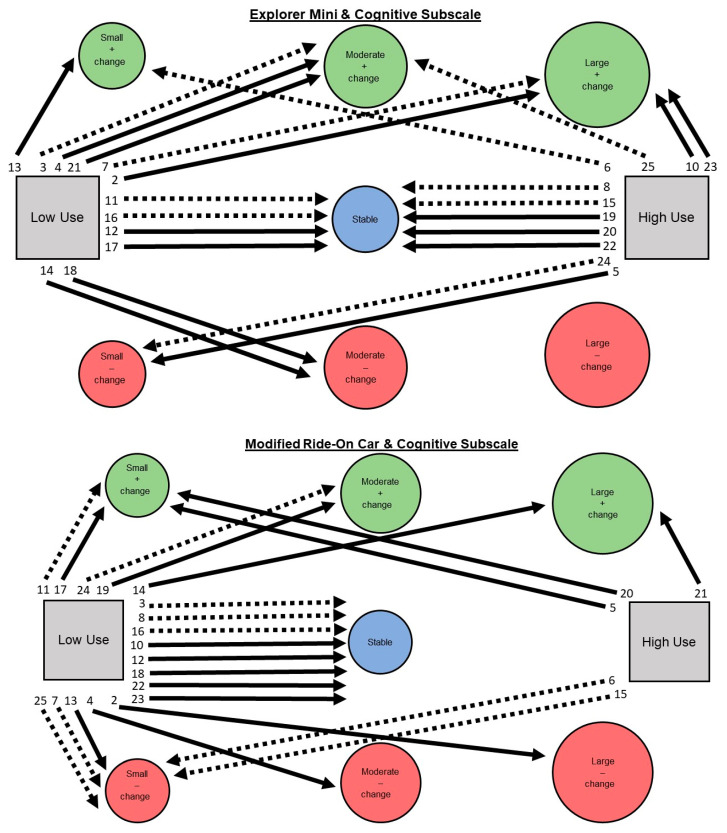
Pathways for each child from Explorer Mini (**top**) and modified ride-on car (**bottom**) low and high use groups to developmental change on the cognitive subscale of the Bayley-4. Low use was defined as 480 min or less across an 8-week period. High use was defined as 481 min or more across an 8-week period. The magnitude of the percentage of change for Bayley-4 subscales was defined as follows: stable (+/−9% or less), small change (+/−10–19%), moderate change (+/−20–29%), or large change (+/−30% or more). The #s indicate participant ID. Bolded lines represent children GMFCS Levels IV or V. Dashed lines represent children GMFCS Levels I, II, or III.

**Figure 4 behavsci-13-00399-f004:**
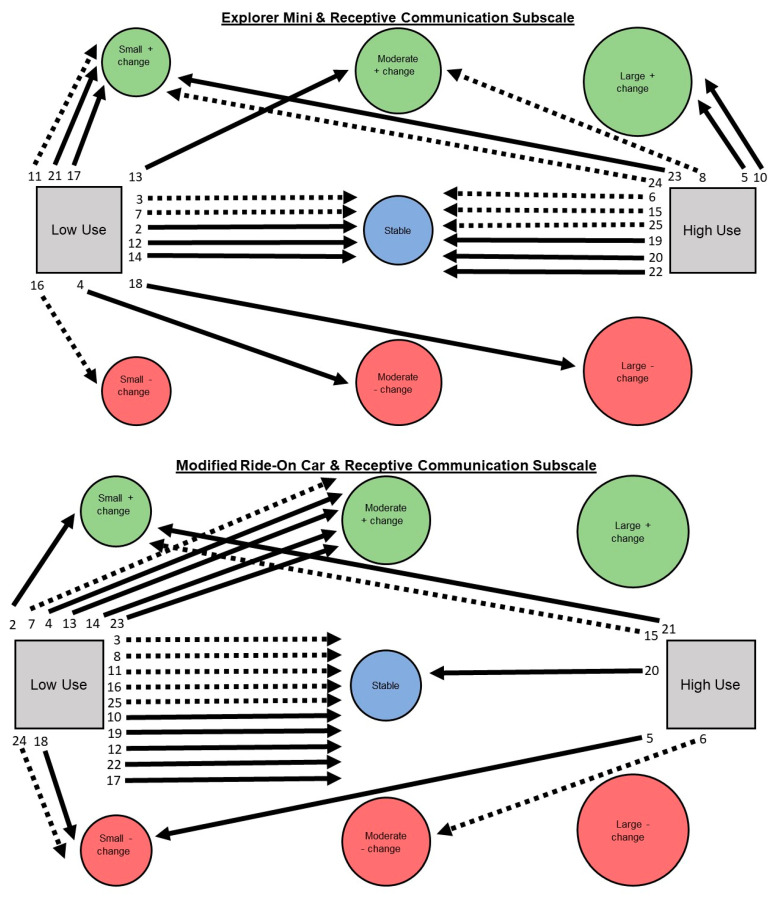
Pathways for each child from Explorer Mini (**top**) and modified ride-on car (**bottom**) low and high use groups to developmental change on the receptive communication subscale of the Bayley-4.

**Figure 5 behavsci-13-00399-f005:**
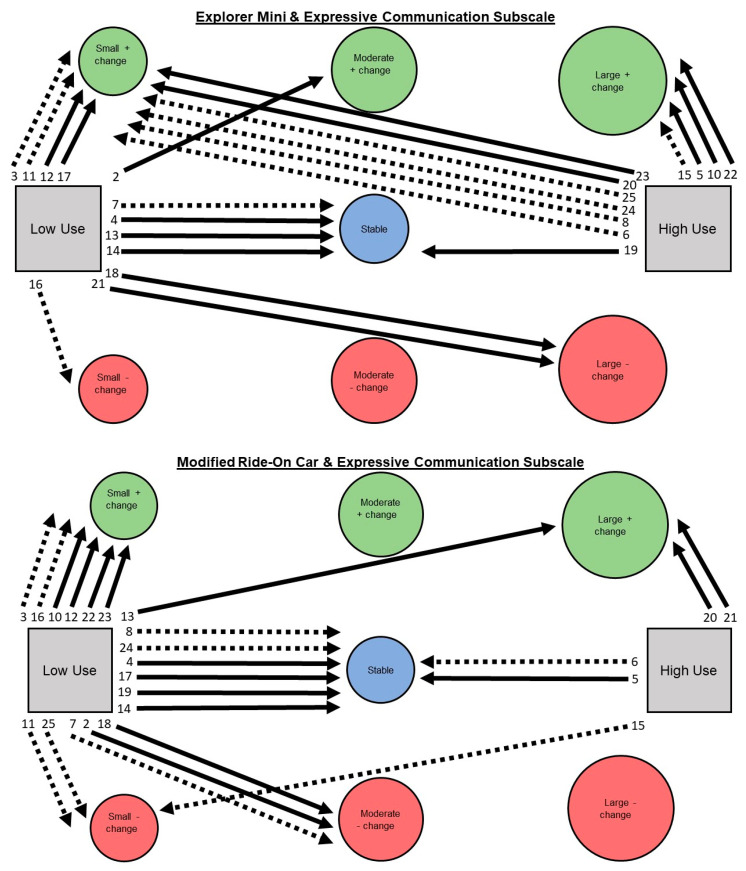
Pathways for each child from Explorer Mini (**top**) and modified ride-on car (**bottom**) low and high use groups to developmental change on the expressive communication subscale of the Bayley-4.

**Figure 6 behavsci-13-00399-f006:**
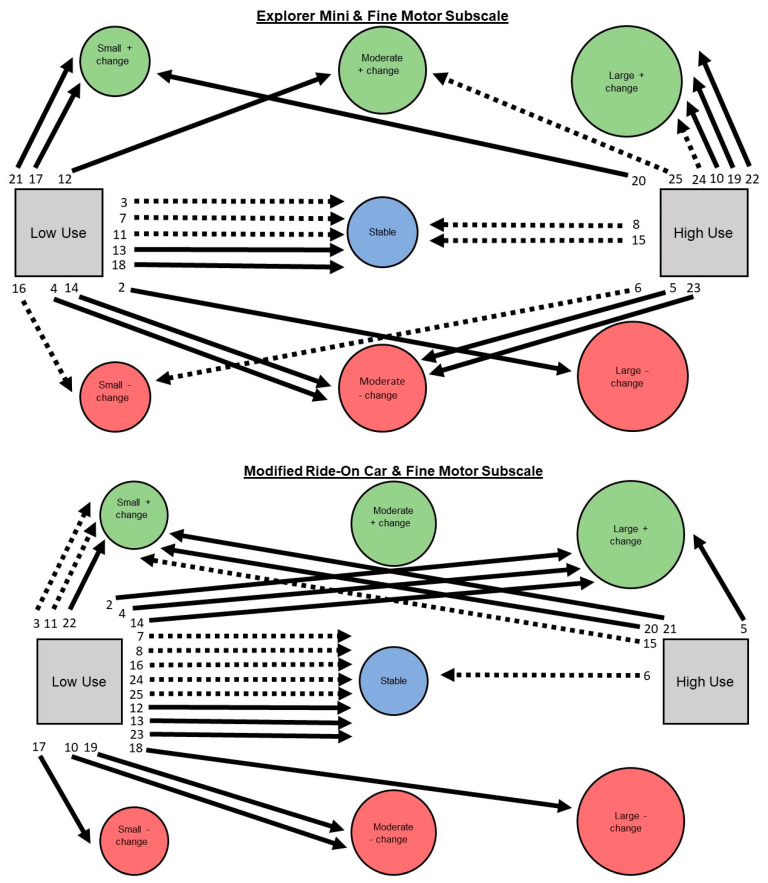
Pathways for each child from Explorer Mini (**top**) and modified ride-on car (**bottom**) low and high use groups to developmental change on the fine motor subscale of the Bayley-4.

**Figure 7 behavsci-13-00399-f007:**
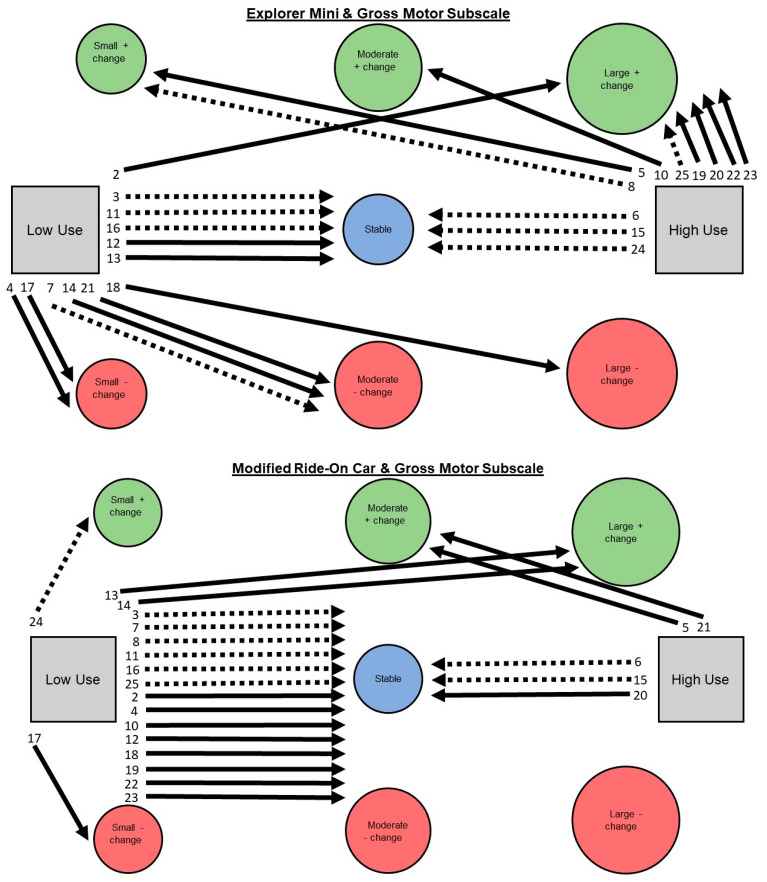
Pathways for each child from Explorer Mini (**top**) and modified ride-on car (**bottom**) low and high use groups to developmental change on the gross motor subscale of the Bayley-4.

**Table 1 behavsci-13-00399-t001:** Demographic information for participants and individual device use data.

			Device Use (Mins)	
ID	Age	GMFCS Level	Explorer Mini	Modified Ride-On Car
2	17 months	V	182	203
3	2 years	II	435	95
4	19 months	V	374	370
5	18 months	V	1185	526
6	2 years and 4 months	II	980	923
7	21 months	III	165	15
8	2 years and 5 months	III	563	48
10	15 months	V	505	86
11	17 months	II	177	99
12	15 months	IV	335	0
13	2 years and 6 months	V	35	60
14	12 months	IV	330	200
15	2 years and 5 months	III	1270	547
16	21 months	I	17	92
17	2 years and 7 months	V	165	30
18	20 months	V	206	0
19	2 years and 5 months	IV	613	0
20	23 months	IV	613	529
21	23 months	V	217	571
22	16 months	IV	1105	119
23	2 years 8 months	V	823	96
24	12 months	II	1110	230
25	18 months	II	547	165

**Table 2 behavsci-13-00399-t002:** Summary information about device use.

	Explorer Mini		Modified Ride-On Car	
	Minutes of Use		Minutes of Use	
	Low Use (*n* = 12)	High Use (*n* = 11)	Low Use (*n* = 18)	High Use (*n* = 5)
**Min**	17	505	0	526
**Max**	435	1270	370	923
**Median**	193.8	823	93.5	538
**Mean**	219.8	846.7	106.4	631.3
**Standard Deviation**	128	290.6	96.9	194.7

**Table 3 behavsci-13-00399-t003:** Frequency and percentages of paths from low and high use to developmental change for each device on the Bayley-4 (all subscales).

Explorer Mini					
**Low Use (*n* = 12; 60 paths)**			**High Use (*n* = 11; 55 paths)**		
(+) Change	**Stable**	(−) Change	**(+) Change**	Stable	(−) Change
19 (32%)	**23 (38%)**	18 (30%)	**33 (60%)**	17 (31%)	5 (9%)
**Modified Ride-On Car**					
**Low Use (*n* = 18; 90 paths)**			**High Use (*n* = 5; 25 paths)**		
(+) Change	**Stable**	(−) Change	**(+) Change**	Stable	(−) Change
27 (30%)	**46 (51%)**	17 (19%)	**13 (52%)**	7 (28%)	5 (20%)

## Data Availability

De-identified individual participant data, including score report data from participation and developmental measures and device use reports, will be made available to other researchers and reported on clinicaltrials.gov per NIH funding requirements and will be made available 6 months following publication of study results. This study has been registered with the US National Library of Medicine Clinical Trial Registry under the National Clinical Trial (NCT) identified number NCT04684576 (Protocol Version 1) on 24 December 2020.

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
