# Peer review of "Powered Mobility Device Use and Developmental Change of Young Children with Cerebral Palsy"

_behavsci, 2023, doi:10.3390/bs13050399_

Round 1
Reviewer 1 Report
Thank you for the opportunity to review this interesting paper. My comments and suggestions are listed in the attached document. Thank you.

Reviewer 2 Report
This study examines the use of powered mobility devices and developmental changes in young children with cerebral palsy.
The study is well presented, and the focus and outcomes are crystal obvious.
However, greater focus should be placed on the mechanics of cutting-edge gadgets and the one chosen to increase the credibility of the work by explaining the benefits and drawbacks of the chosen methodology. A basic section has been created, however it should be more detailed.
Using and mentioning these articles can help with this:
1.https://doi.org/10.1007/978-3-319-09858-6_47
2. https://doi.org/10.3390/cryst11050561
Finally, additional photographs should be provided (e.g. device, etc.). To provide the reader with a better understanding.
Round 2
Reviewer 1 Report
I am satisfied that the authors have addressed all my comments and concerns on the manuscript. I have a single suggestion for the authors to correct a typo on Page 17, Line 542 where the authors should correct their included age range to "12-36 months" instead of "12-32 months"I would recommend the publication of the manuscript in your journal once the authors make this correction.
Author Response
Thank you for the comment. We have updated "32" to "36". Thanks again!